# *Ligilactobacillus salivarius* MP100 as an Alternative to Metaphylactic Antimicrobials in Swine: The Impact on Production Parameters and Meat Composition

**DOI:** 10.3390/ani13101653

**Published:** 2023-05-16

**Authors:** Claudio Alba, David Castejón, Víctor Remiro, Juan M. Rodríguez, Odón J. Sobrino, Julián de María, Pilar Fumanal, Antonio Fumanal, M. Isabel Cambero

**Affiliations:** 1Department Nutrition and Food Science, Complutense University of Madrid, 28040 Madrid, Spain; 2ICTS Bioimagen Complutense (BIOIMAC), Universidad Complutense de Madrid. Pº de Juan XXIII 1, 28040 Madrid, Spain; 3Department Galenic Pharmacy and Food Technology, Complutense University of Madrid, 28040 Madrid, Spain; 4Scientific Society of Veterinary Public and Community Health (SOCIVESC), 28040 Madrid, Spain; 5L’Albeitar, 22349 Huesca, Spain

**Keywords:** *Ligilactobacillus salivarius*, swine, probiotics, growth performance, meat quality

## Abstract

**Simple Summary:**

Antibiotic usage in swine husbandry contributes to the emergence and dissemination of antibiotic-resistant bacteria, posing a serious threat to public health. Consequently, alternative strategies are required to mitigate this issue. In a previous study, we showed that the replacement of metaphylactic antimicrobials with an administration of *Ligilactobacillus salivarius* MP100 for two years to sows and piglets changed the fecal microbiota and short-chain fatty acid profiles in an intensive pig farm. In this work, we compared the productivity-related data between a 2-year period of routine metaphylactic antibiotherapy and the first 2 years in which such a practice was replaced by the probiotic strain. Subsequently, the composition of the meat of the animals ingesting the probiotic strain was compared to that of animals with a similar diet but receiving metaphylactic antibiotherapy. This probiotic intake did not negatively alter the composition of the meat and was associated with an increase in the concentration of inosine and a slight tendency for increasing the thickness of the subcutaneour lumbar fat and intramuscular fat content. These factors are considered as biomarkers of meat quality and health. Therefore, the substitution of metaphylactic antimicrobials with a daily administration of the probiotic strain was associated with beneficial productivity and quality outcomes.

**Abstract:**

The metaphylactic use of antimicrobials in swine farms contributes to the emergence of antibiotic-resistant bacteria, which constitutes a major challenge for public health. Alternative strategies are required to eradicate their routine use. In a previous study, metaphylactic antimicrobials were replaced by the administration of *Ligilactobacillus salivarius* MP100 to sows and piglets for two years. This practice positively modified the fecal microbiota and metabolic profiles in the farm. In this work, the farm dataset was used to compare the productivity-related parameters between a 2-year period of routine metaphylactic antibiotherapy and the first 2 years of a replacement with the probiotic strain. The probiotic period improved these productivity-related parameters, from litter size to growth performance. In addition, samples of *Longissimus lumborum*, including skin and subcutaneous fat, were obtained from the animals ingesting the probiotic strain and controls (metaphylactic antibiotherapy) and analyzed for their pH, water holding capacity, composition, and metabolic profiling. The probiotic intake did not negatively affect the meat composition and was associated with an increase in inosine concentration and a slight tendency for increasing the intramuscular fat content. These factors are considered as biomarkers of meat quality. In conclusion, the substitution of metaphylactic antimicrobials with the administration of the probiotic strain was associated with beneficial productivity and meat quality outcomes.

## 1. Introduction

Over the last decades, the widespread use of antibiotics in animals and humans has driven the fast emergence and spread of antibiotic-(multi)resistant bacteria. This situation has been considered as one of the most relevant threats to animal and human health in the near future [1,2,3]. In this context, the routine prophylactic or metaphylactic use of antibiotics in intensive animal systems, and particularly in swine production, seems especially worrying [4,5]. The use of some alternatives to eradicate or minimize this prophylactic or metaphylactic antibiotherapy in pig production have been proposed and they include acidifiers, prebiotics, and probiotics [6,7]. 

In relation to probiotics, it has been reported that the administration of some lactobacilli strains may provide beneficial outcomes for performance [8,9,10,11] and meat quality [11,12,13]; however, only a few strains have been tested in real farm conditions. In addition, studies with other probiotic strains, mainly belonging to the genus *Bacillus*, have failed to find a positive impact or have even reported a detrimental effect on such parameters [14]. Such conflicting results may reflect the differences in the properties of the administered strains and, therefore, a proper characterization of the efficacy- and safety-related traits has a paramount relevance when selecting the probiotic strains targeting swine production.

In this context, we have recently described the probiotic potential of a *Ligilactobacillus salivarius* strain isolated from sow’s milk [15]. In the same study, the strain was routinely administered (daily for two years) to pregnant sows and piglets as an inoculated, fermented feed. The same day that the probiotic intervention started, the metaphylactic use of antimicrobials (including zinc oxide), which was the routine practice of the farm until that moment, was completely eliminated. The probiotic intervention was associated with higher abundances of clostridia and lactobacilli in the feces of 8-week-old piglets, higher fecal concentrations of short-chain fatty acids (acetate, propionate, and butyrate), and a significant decrease in the rates of antibiotic-resistant lactobacilli [15]. Overall, this change in farm management provided beneficial outcomes for the intestinal health of the animals, but also for public health. 

Since the farm kept a complete historic register of the productivity parameters (mortality rates at weaning and fattening, total fattening days, daily weight increases, and number of liveborn piglets per sow), the first objective of this work was to elucidate if the replacement of antimicrobials with a microbiota-friendly approach was able to provide benefits for these production-related parameters and, therefore, to elucidate if such an approach is economically viable and sustainable. For this purpose, long 2-year periods were compared to investigate the effects of probiotic treatment and the use of antibiotics in pig farming. The first period involved routine metaphylactic antibiotherapy, while the second period was the first two years of antibiotherapy replacement with the probiotic strain.

The second objective of this work was to characterize the composition of pork meat after 2 years of probiotic consumption. For this purpose, the meat of the commercial animals reared on this farm was compared with the meat of animals reared on another farm, which used the same feed, but also kept the routine use of metaphylactic antibiotherapy. 

## 2. Materials and Methods

### 2.1. Farm Description and Study Design

The study was conducted on an intensive swine farm (L’Albeitar, Guaso, Huesca, Spain), a closed-cycle pig farm with a farrow-to-finish herd consisting of 210 genetically similar Large White × Landrace sows. Between 40 and 50 of all the sows born at the farm each year are kept for the replacement of reproductive sows. Each reproductive sow is kept for 7–8 pregnancies. The sows are artificially inseminated with semen from TN Tempo Large White boars (AIM, Vught, The Netherlands) and have been since 2013. The main characteristics of the farm facilities and its management practices have been described previously [15]. The design of the 2-year probiotic intervention, concomitant with the eradication of the metaphylactic use of antibiotics, as well as the characteristics of the routine metaphylactic use of antimicrobials before the start of the probiotic trial, were also described in the same article. Briefly, during the probiotic trial, the animals received the same diet as previous, with the exception of all the antimicrobials (antibiotics and zinc oxide) being removed from their feed. The strain *Ligilactobacillus salivarius* MP100 was administered daily (∼9 log_10_ colony-forming units [CFU]) each day) for 2 years as a fermented feed to pregnant sows (from the week before farrowing to the end of lactation) and piglets (from 12 days after birth to the initiation of the fattening period). The study was approved by the Ethical Committee on Animal Experimentation of the Faculty of Veterinary Sciences of the Universidad Complutense de Madrid (Spain), under protocol number 33/17.

### 2.2. Production-Related Parameters

The farm keeps a complete historic register of its productivity parameters, which were measured by the same technicians and used the same procedures and criteria for all the sow and piglet batches reared on the farm for at least the past 10 years. The parameters analyzed in this work included the fertility rate (the percentage of inseminated sows that had a litter), average litter size at birth, piglet mortality rates during the nursery, transition, and fattening stages, total mortality rates, average weight of pigs at slaughter, number of days from birth to slaughter, and average daily weight gains. To measure these daily weight gains, the weight of each animal at slaughter was divided by the number of days from its birth. This provided a measure of the average daily weight gain, which is an effective way of assessing an animal’s growth and development over time. To evaluate the impact of the probiotic administration and eradication of antimicrobial prophylaxis, the data obtained over two well separated 2-year periods were compared: (a) 2014–2015 (routine metaphylactic antibiotherapy, no probiotic), and (b) 2018–2019 (the first two years of antimicrobial replacement with the probiotic strain). The intervening years, 2016–2017, were characterized by a gradual reduction in antibiotic usage, unaccompanied by probiotic supplementation.

### 2.3. Characterization and Composition of Meat and Subcutaneous Fat

To address the question of whether the 2-year administration of the probiotic strain in the farm could have affected the quality of meat and subcutaneous fat, a metabolic profiling study was conducted. For this purpose, ten commercial castrated male pigs from the study farm were compared to ten individuals from a control farm that used the same feed, but also a standard metaphylactic antimicrobial treatment. The feed was elaborated in Cooperativa Alto Aragón de Barbastro SCL (Barbastro, Spain) and contained wheat, barley, corn, bran soybeans, beet pulp, lard (3.9%), and vitamin and mineral correctors. This proportion of lard remains unchanged over time. In no case were by-products, fats, or oils of any other type used, only noble raw materials, and the feed was always made using the same formula and technology in a feed factory in the area. All the animals were randomly selected in the slaughter house, where the animals were sacrificed (Fribin, Binéfar, Spain). A slice of *Longissimus lumborum* (LL), including skin and associated subcutaneous fat, was taken from the end of the last rib. These portions had total thicknesses of 3 to 5 cm (at least 200 g) and were taken 24 h post-slaughter, from the left half carcass of the animals. The subcutaneous fat thickness was measured (without including the skin) at 6 cm of the dorsal midline using a caliper (Digit Cal, Tesa, Brown and Sharpe, Renens, Switzerland). Then, the subcutaneous fat lawyer was separated from the muscle and both parts were submitted for their respective analyses (see below). 

Initially, some technological characterization parameters, such as pH, water holding capacity (WHC), composition (including fatty acid profile), and metabolic profiling, were determined. Subsequently, and to complete the characterization study of the pork samples, an analysis of their metabolites using ^1^H NMR spectroscopy was carried out. For this purpose, the released exudate from the packaged LL portions was collected and analyzed, since the meat exudate is an excellent alternative analytical matrix for providing complete and homogeneous metabolic information about the whole meat piece [16,17].

### 2.4. Physicochemical Analyses of LL Samples

The pH was determined in a homogenate of the sample with distilled water (1:10) (*w*/*v*), using a Crison Digit-501 pH meter (Crison Instruments LTD, Barcelona, Spain). The water content was analyzed following the oven air-drying procedure, while the protein (Kjeldhal nitrogen) content was determined using the standard AOAC method [18]. The fat content was determined using the method of Bligh and Dyer, as described by Hanson and Olley [19].

The water holding capacity (WHC) was measured by using the Carver Press Method [20,21]. The meat sample (0.3 g) was placed on a piece of filter paper (Whatman no. 1, 125 mm), then set between two plexiglas plates, and subjected to a mechanical force of 345 kPa for 5 min. The WHC values were calculated as the percentage of water retained based on the water content in the product before the pressing. Four replicates of each sample were determined.

### 2.5. Fatty Acid Profile Analysis of LL Intramuscular Fat and Subcutaneous Fat

For the fatty acid analysis, fatty acid methyl esters (FAMEs) were obtained from isolated lipids by heating the samples at 80 °C for 1 h in 3 mL of methanol/toluene/ H_2_SO_4_ (88:10:2 by volume) [22]. After cooling, 1 mL of hexane was added and the samples were mixed. The fatty acid methyl esters were recovered from the upper phase, separated, and quantified using a gas chromatograph (HP 6890 Series GC System) equipped with a flame ionization detector. The separation was performed with a J&W GC HP-Innowax Polyethylene Glycol (30 m × 0.316 mm × 0.25 μm) column (Agilent, Santa Clara, CA, USA). Nitrogen was used as a carrier gas. The fatty acids analysis was carried out as described in Segura et al. [23].

### 2.6. Meat Metabolites Analysis by ^1^H NMR Spectroscopy

The released exudate, obtained after 48 h postmortem from either the probiotic group samples (10 animals) or control group animals (10 animals), was vacuum-packed, frozen at −80 °C, and freeze-dried at room temperature. The freeze-dried exudates were stored at −80 °C until the NMR analysis.

For the ^1^H NMR analysis, 18 mg of the lyophilized pork tenderloin (LL) exudates was reconstituted in an eppendorf by adding 650 μL of deuterium oxide (D_2_O, VWR). The samples were vortexed for 30 s and transferred into 5 mm NMR tubes.

The ^1^H NMR spectra of the reconstituted exudates were recorded randomly at 298 K in a Bruker Avance 500 MHz spectrometer using a 5 mm multinuclear direct detection broadband probe (Bruker BioSpin GmbH, Rheinstetten, Germany). For each sample, two monodimensional ^1^H NMR spectra were recorded using the following pulse sequences: 1D NOESY (NOESYPRESAT) and CPMG (Carr Purcell Meiboom Gill), both with water suppression. The 1D NMR experiments were acquired with 32 k/73 k time domain data points, an 18/24 ppm spectral width, 64 scans, and 2/4 s of relaxation delay. A mixing time of 150 ms was used for the 1D NOESY experiment and an echo time of 50 ms was selected for the CPMG sequence. The free induction decays (FIDs) obtained were processed with the Bruker BioSpin TOPSPIN software (version 4.1.4). Prior to Fourier transformation, the FIDs were multiplied by an exponential weight function corresponding to a line broadening of 0.3 Hz. The spectra were phased, baseline-corrected, and referenced to the alanine doublet at δ = 1.485 ppm. The assignment of the resonances in the ^1^H NMR spectra from the pork tenderloin exudates was based on previous studies [16,24].

### 2.7. Statistical Analysis

The normality of all the variables was assessed using the Shapiro–Wilk test with the “stats” package in R (R citation) and the data are reported as means ± 95% confidence intervals (±CI) or means and standard deviations (SD). As all the variables followed a normal distribution, the student *t*-test was used to detect any significant differences between the antibiotic and probiotic consumption batches.

## 3. Results

### 3.1. Fertility Rate and Litter Size

The total number of inseminated sows in the metaphylactic antimicrobial period was 1009, while it was 1385 in the probiotic period. Among them, 85.99% (±2.47) and 87.45% (±1.67) of the sows gave birth to live litters in the respective periods. There were no statistically significant differences between the two periods in relation to the fertility rates (*p* = 0.34) (Figure 1a).

The average litter size (the number of live piglets per litter) in the metaphylactic antimicrobial period was 12.62 (±0.30), while it increased to 13.15 (±0.17) in the probiotic period. Such an increase was statistically significant (*p* = 0.003). Among the litters, the numbers of piglets that were eventually weaned were 10.98 (±0.24) and 11.33 (±0.20) in the respective periods and this difference was statistically significant too (*p* = 0.03) (Figure 1b).

### 3.2. Mortality Rates during the Nursing, Post-Weaning, and Fattening Stages

The mortality rate among the nursing piglets was lower during the metaphylactic antimicrobial period than that during the probiotic period (12.89 ± 1.30% vs. 13.86 ± 0.71%, respectively); however, this difference was not statistically significant (*p* = 0.19). The mortality rates during the post-weaning stage were 1.11% (±0.25) and 2.18% (±0.56), respectively (*p* = 0.004). Finally, the mortality rates during the fattening stage did not show a statistically significant difference when both periods were compared (6.58% [±0.84] and 6.30% [±0.68], respectively; *p* = 0.62). Overall, there were no statistically significant differences in the total piglet mortality rates (i.e., the mortality rate from birth to slaughter) when both periods were compared (19.13 ± 1.72% and 20.48 ± 1.16%, respectively; *p* = 0.20) (Figure 2).

### 3.3. Number of Days from Birth to Slaughter, Average Weight of Pigs in the Day of Slaughter, and the Average Daily Weight Gain (g)

The number of days until slaughter was significantly higher during the metaphylactic antimicrobial period than that during the probiotic period (171.84 ± 1.40 and 165.57 ± 1.05 days, respectively; *p* < 0.001) (Figure 3a). The mean weight of the pigs on the day of slaughter was significantly lower during the metaphylactic antimicrobial period than that during the probiotic period (108.88 kg [±0.83] and 110.51 kg [±1.05]; *p* = 0.03). Similarly, the average weight gain per day was significantly lower during the metaphylactic antimicrobial period than that during the probiotic period (625.32 g ± 7.54 and 658.74 g ± 8.00, respectively; *p* < 0.001) (Figure 3b). 

### 3.4. Physicochemical Analyses of LL Samples

The physicochemical characteristics of the LL meat obtained from the animals of the probiotic and control groups, as well as the thicknesses of the subcutaneous lumbar fat measured at the level of the last rib, are shown in Table 1. The characteristics of the samples of both groups were very similar, differing only in their intramuscular fat content, which was higher when the meat came from the animals of the probiotic group (*p* > 0.05), coinciding with their higher growth rate. In the same line, the probiotic-fed animals presented greater subcutaneous fat thicknesses (*p* < 0.05).

### 3.5. Fatty Acid Profile Analysis of LL Intramuscular Fat and Subcutaneous Fat

To characterize the meat and subcutaneous fat, the fatty acid profiles were also determined (Table 2). Similar to the previous parameters, no significant quantitative differences were detected between the samples from the two groups for any of the individual fatty acids or for the proportion of a specific type of fatty acid (SFA, MUFA, PUFA, n-6, n-3, and hyper- and hypo-cholesterolemic) (*p* > 0.05). In general, the intramuscular fat (IMF) of the LL and subcutaneous fat was characterized by a high proportion of monounsaturated fatty acids (MUFA, 44–48%). The proportion of saturated fatty acids (SFA) was around 35–36% and the polyunsaturated fatty acids (PUFA) content ranked around 17–20% (Table 2). In relation to the individual fatty acids, the IMF and subcutaneous fat showed higher percentages of C18:1n-9 (37–38% in IMF and 39–42% in subcutaneous fat), C16:0 (about 22% in both fats), C18:2n-6 (about 15% in both fats), and C18:0 (about 11–12%) (Table 2).

### 3.6. Comparative Study of the ^1^H NMR Spectra of Samples from Animals of the Probiotic and Control Groups

A comparison between the ^1^H NMR spectra of the samples obtained from the animals of the probiotic and control groups demonstrated that both groups displayed similar profiles. Figure 4 presents a representative 1 H NMR spectrum of the LL exudates from the animals of the probiotic group, while Appendix A shows a list of all the metabolites identified in the NMR spectra and their spectroscopic data. A total of 50 metabolites could be identified through the analysis of the exudate spectra, including amino acids, peptides (such as carnosine and anserine) and analogues, carbohydrates, lactate, nucleotides, creatine, fatty acids, and organic acids. 

Despite the great similarity between the LL exudate spectra obtained from the samples of both groups of pigs, a more detailed analysis of the data revealed differences in some regions of the spectra, especially in those corresponding to the inosine signals (6.08–6.12 ppm). More specifically, the presence of inosine was detected in all the samples from the pigs in the probiotic group, while it was not detected in the control group or was found in much lower concentrations than those detected in the probiotic group. Such a specific region has been expanded (within a box) in Figure 1. 

## 4. Discussion

The results of this study reveal that the replacement of metaphylactic antimicrobial practices with the administration of a well-characterized swine-targeted probiotic strain led to an overall improvement in productivity-related parameters. Although there were no significant differences between the two periods in relation to the fertility or total mortality rates, the average litter size and number of piglets that were eventually weaned were significantly higher during the probiotic period. In addition, the number of days to reach the commercial age for slaughter was significantly lower during the probiotic period, while the mean weight of the pigs on the day of slaughter and average weight gain per day were significantly higher in the same period when compared to the metaphylactic antimicrobial period. Therefore, the substitution of metaphylactic antimicrobials with the daily administration of a selected probiotic strain had a positive impact on farm productivity in practice. It must be highlighted that no major changes have been introduced to this farm in relation to housing conditions, genetics, or nutrition since 2013; therefore, the impact of these factors when comparing the two 2-year periods was probably low. In relation to the weather conditions, although some differences were observed among the different months and years, the mean temperatures and pluviometry trimestral values were very similar when both 2-year periods were compared (SiCLIMA; Sistema Básico de Información Climática de Aragón). More specifically, the mean daily temperature values in the antibiotic period and probiotic period in the farm location were, respectively: 11.2 and 11.5 °C in spring; 20.2 and 20.5 °C in summer; 14.3 and 14.1 °C in autumn; and 5.5 and 5.3 °C in winter. In relation to pluviometry, the mean daily pluviometry values were, respectively, as follows: 150.4 and 151.3 mm in spring; 91.1 and 87.4 mm in summer; 164.9 and 160.2 mm in autumn; and 115.3 and 118.7 mm in winter.

In agreement with the results of this work, previous studies have also demonstrated that that an oral supplementation with *Lactobacillus*-based probiotics improves the growth performance, including the average daily gain and gain-to-feed ratio in piglets [8]. The microbiota and metabolic shifts associated with the administration of *L. salivarius* MP100 [15] may help in restoring the gut ecosystem, which plays a key role the proper growth and development of pigs [25]. Similar to *L. salivarius* MP100, other pig-origin lactic acid bacteria strains specifically selected for their use in swine, such as *Pediococcus acidilactici* FT28, have been shown to improve this growth performance, including the average daily gain, feeding conversion rates, and/or weight in suckling piglets [9], weaned piglets [10], and growing pigs [11]. In addition, some strains may improve sows’ performance during pregnancy and lactation, a fact that can lead to beneficial outcomes for piglets during growth [12]. All these studies provide evidence of the growth-promoting effects of supplementation with well-selected lactic acid bacteria in swine production and confirm the potential of some probiotic strains as alternatives to antimicrobials in pig production [8,12]. Some *Bacillus* spp. have also been tried as swine growth promoters, but the results seem to depend on the strain. Thus, while one *Bacillus* strain was able to increase the number of born-alive and weaned piglets when administered to sows and piglets [26], supplementation with *Bacillus subtilis* C-3102 during gestation, lactation, and nursery did not improve the litter performance in lactation [14]. On the contrary, the pigs born from the probiotic-fed sows had a lower average daily gain, average daily feed intake, and body weight in the late nursery period than the pigs born from the control sows [14]. These conflicting results highlight the need for a proper selection of strains targeting sows or piglets and the need for trials to assess their safety and efficacy.

In addition to production parameters, the results of some studies suggest that the use of probiotics in pigs may affect the different physicochemical parameters of the derived meat in relation to the meat quality and technological aptitude, such as pH, water holding capacity, or intramuscular fat [11,12,13]. These attributes are important in the perception of meat quality, since they determine the weight loss due to drip loss, which ranges between 2 and 10% [27], and are related to meat juiciness and tenderness perception [28]. In this work, the physicochemical characteristics of the LL samples obtained by both groups (probiotic group and control group) were similar for most parameters.

In relation to the fatty acid profiling of the LL intramuscular fat and subcutaneous fat, no significant differences were detected between the samples of the two groups. This result was expected, since all the pigs were fed with a similar commercial feed and had similar genetics. The fatty acid compositions of pig muscles and adipose tissues are influenced by several factors, but among them, the composition of the dietary fat and/or oil has a paramount influence on the fatty acid content of pork fat [29]. The fatty acid profiles observed in this work were similar to those described in the carcass fat from white pigs fed with commercial diets and were the ranges of the values associated with an adequate consistency of the meat fat, allowing for its cutting and handling in the meat industry for the preparation of different meat products [30]. Regarding health recommendations, the PUFA/SFA ratio of the studied fat was approximately 0.5–0.6 in the intramuscular fat and around 0.4–0.5 in the subcutaneous fat, staying within the recommendations of Ulbricht and Southgate [31], who suggested that the PUFA/SFA ratio should be at least 0.4. The content of the n-6 PUFA was high, maintaining the n-6/n-3 PUFA ratio above 4:1, in accordance with the recommendations of Scollan et al. [32]. The hypercholesterolemic/hypocholesterolemic fatty acids ratio (atherogenic index) was lower than 0.5 (between 0.37 and 0.4), and therefore within the range found by other authors for pigs [30].

In this work, the ^1^H NMR spectroscopy of the meat exudates was used for a global analysis of the metabolites present in the meat. The NMR spectra of complex mixtures show hundreds of signals coming from a great number of diverse metabolites, which, through the adequate treatment of data using chemometrics, allows for the classification and differentiation of samples. In previous works, it has been verified that exudates extracted from beef or pork meat are a suitable matrix for studying meat metabolites and the metabolic changes that occur in the meat during storage and conservation [16,17]. Thus, ^1^H NMR would be a suitable technology for a non-destructive analysis allowing for the monitoring of the different procedures associated with meat processing. In the present work, this technology was used to elucidate the potential effect of probiotic intake on the different components of pork. All the metabolites identified in the pork exudate spectra obtained in this work had been previously described in studies carried out on different meat products [16,17,24]. Overall, the spectra from both groups of animals (probiotic group and control group) were very similar. However, the presence of inosine was clearly detected in all the exudate samples corresponding to the probiotic group, while in the control group samples, it was only detected in some cases and in noticeably lower concentrations. This difference may be relevant since inosine is considered as a biomarker for determining meat quality [33], being detected mainly in meats (such as beef) with higher quality and flavor development [34]. Additionally, the presence of inosine may be relevant for health as well, since several beneficial health effects associated with inosine have been reported [35,36,37].

The statistical analysis of the high amount of complex data obtained in this study was a challenging issue because of some of the intrinsic characteristics of the assay, including the recurrence of swine births over time and the potential occurrence of random effects. Although advanced statistical models, such as mixed models or mixed effects models, are suitable in such scenarios, our approach relied on the use of *t*-tests to compare the data obtained during both 2-year periods under similar meteorological and management conditions, with the only exception being the replacement of antimicrobials with a probiotic strain. Although we tried to keep the variability at a minimum, the occurrence of random factors affecting the interpretation of our data cannot be excluded and this is one of the main limitations of this study. More studies are required to confirm the benefits of a routine administration of *L. salivarius* MP100 to sows and piglets, including farms with different management systems and/or in different environmental settings. In addition, a large number of samples will be required to further assess the impact of the strain on the meat composition. 

## 5. Conclusions

The replacement of metaphylactic antimicrobial practices with an administration of *L. salivarius* MP100 to sows and piglets improved their productivity-related parameters, including the average litter size, number of weaned piglets, number of days to reach the commercial slaughter age, mean weight of the pigs on the day of slaughter, and average weight gain per day. Therefore, the substitution of metaphylactic antimicrobials with this daily administration of the probiotic strain was feasible and economically affordable. In addition, supplementation with the strain did not alter the composition of the pork meat and was associated with an increase in the concentration of inosine, which is considered as a biomarker of meat quality and health. The probiotic intake was also related to an increase in the thickness of the subcutaneous lumbar fat and intramuscular fat contents of *Longissimus lumborum*. These factors would contribute to the juiciness and flavor development of the meat.

In conclusion, this strain is a promising tool for reducing or eliminating the metaphylactic use of antimicrobials in swine production, a fact that must be confirmed in future well-designed trials involving a higher number of animals and samples.

## Figures and Tables

**Figure 1 animals-13-01653-f001:**
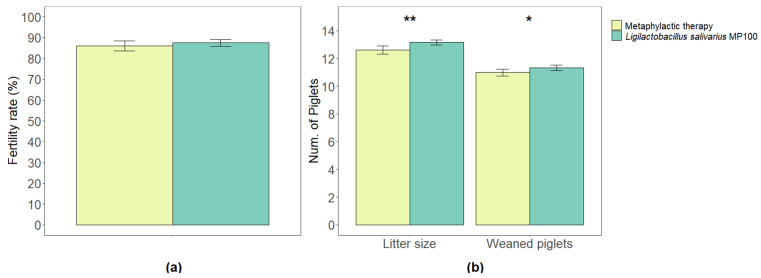
Comparison of fertility rates (% of inseminated sows that gave birth to a live litter) (**a**), and average litter size and final number of piglets weaned per sow (**b**) between the two periods analyzed in this study (metaphylactic antimicrobial period versus probiotic period). Significant differences are indicated as * (*p* < 0.05) or ** (*p* < 0.01).

**Figure 2 animals-13-01653-f002:**
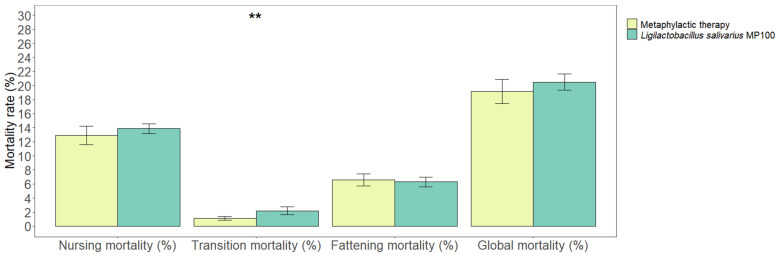
Comparison of mortality rates (%) in the nursing, transition, and fattening stages, and overall mortality rates from birth to sale between the two periods analyzed in this study (metaphylactic antimicrobial period versus probiotic period). Significant differences are indicated as ** (*p* < 0.01).

**Figure 3 animals-13-01653-f003:**
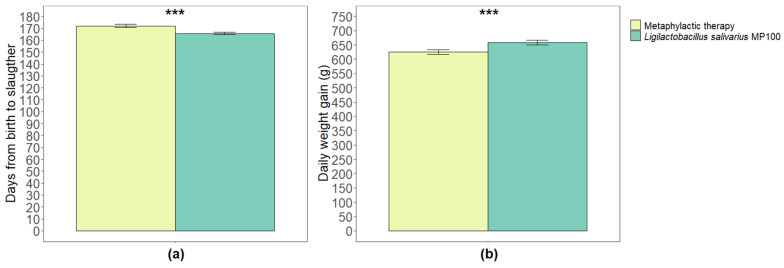
Comparison of average number of days from birth to slaughter (**a**), and average daily weight gain (**b**) between the two periods analyzed in this study (metaphylactic antimicrobial period versus probiotic period). Significant differences are indicated as *** (*p* < 0.001).

**Figure 4 animals-13-01653-f004:**
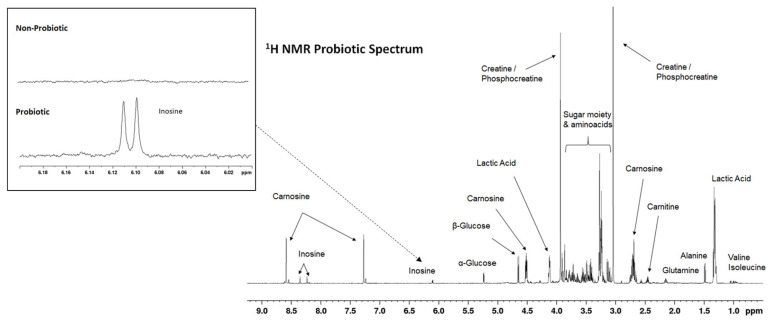
Representative ^1^H NMR spectra of an LL exudate analyzed in this study. Peak assignments are given in Appendix A. The box shows a region (inosine signals) in which differences between the samples of the probiotic and control groups were detected.

**Table 1 animals-13-01653-t001:** Physicochemical characteristics (mean ± SD) of *Longissimus lumborum* muscle and subcutaneous fat.

	Control Group	Probiotic Group
Moisture (%)	72.62 ± 2.22 ^a^	72.88 ± 1.98 ^a^
Ash (%)	1.17 ± 0.16 ^a^	1.10 ± 0.11 ^a^
Protein (%)	22.21 ± 2.06 ^a^	20.93 ± 1.82 ^a^
Intramuscular fat	4.26 ± 0.68 ^b^	5.97 ± 0.57 ^a^
SFT * (mm)	7.75 ± 2.01 ^b^	12.67 ± 2.05 ^a^
WHC **	42.89 ± 4.01 ^a^	42.97 ± 4.47 ^a^
pH (24 h *post mortem*)	5.38 ± 0.10 ^a^	5.48 ± 0.08 ^a^

^a, b^: Value in a row with different letters are significantly different (*p* < 0,05) (*t*-test); equal letters for groups mean no differences for that characteristic. * Subcutaneous fat thickness (SFT) average thickness (not including rind) measured at the level of the last rib (at 6 cm of the dorsal midline). ** Water holding capacity (WHC).

**Table 2 animals-13-01653-t002:** Fatty acid profiles (% of total fatty acids [mean ± SD]) of pork *Longissimus lumborum* muscle and subcutaneous fat.

Batch	Intramuscular Fat		Subcutaneous Fat	
Fatty Acid	Control Group	Probiotic Group	Control Group	Probiotic Group
C14:0	1.18 ± 0.19 ^a^	1.00 ± 0.17 ^a^	1.25 ± 0.09 ^a^	1.17 ± 0.11 ^a^
C14:1	0.13 ± 0.07 ^a^	0.11 ± 0.05 ^a^	0.01 ± 0.00 ^a^	0.01 ± 0.00 ^a^
C15:0	0.08 ± 0.01 ^a^	0.07 ± 0.02 ^a^	0.06 ± 0.01 ^a^	0.05 ± 0.01 ^a^
C16:0	22.25 ± 1.69 ^a^	22.12 ± 0.83 ^a^	22.82 ± 0.81 ^a^	21.88 ± 1.44 ^a^
C16:1n-9	0.43 ± 0.07 ^a^	0.47 ± 0.09 ^a^	0.47 ± 0.08 ^a^	0.47 ± 0.09 ^a^
C16:1n-7	2.39 ± 0.33 ^a^	2.82 ± 0.52 ^a^	2.04 ± 0.26 ^a^	2.21 ± 0.21 ^a^
C17:0	0.31 ± 0.07 ^a^	0.23 ± 0.04 ^a^	0.39 ± 0.06 ^a^	0.33 ± 0.07 ^a^
C17:1	0.37 ± 0.05 ^a^	0.34 ± 0.08 ^a^	0.37 ± 0.05 ^a^	0.32 ± 0.08 ^a^
C18:0	11.34 ± 0.98 ^a^	10.73 ± 0.9 ^a^	12.18 ± 0.98 ^a^	11.28 ± 1.31 ^a^
C18:1n-9	37.85 ± 3.78 ^a^	37.51 ± 3.01 ^a^	39.96 ± 1.94 ^a^	41.76 ± 1.54 ^a^
C18:1n-7	2.86 ± 0.55 ^a^	3.76 ± 0.83 ^a^	2.24 ± 0.47 ^a^	2.40 ± 0.24 ^a^
C18:2n-6	14.94 ± 1.96 ^a^	15.43 ± 1.95 ^a^	15.11 ± 1.16 ^a^	14.96 ± 1.88 ^a^
C18:3n-6	0.12 ± 0.09 ^a^	0.17 ± 0.05 ^a^	0.04 ± 0.02 ^a^	0.02 ± 0.00 ^a^
C18:3n-3	0.71 ± 0.09 ^a^	0.55 ± 0.11 ^a^	0.93 ± 0.11 ^a^	1.07 ± 0.17 ^a^
C18:4n-3	0.09 ± 0.03 ^a^	0.11 ± 0.02 ^a^	0.1 ± 0.02 ^a^	0.08 ± 0.02 ^a^
C20:0	0.17 ± 0.03 ^a^	0.15 ± 0.02 ^a^	0.18 ± 0.04 ^a^	0.17 ± 0.04 ^a^
C20:1n-9	0.62 ± 0.19 ^a^	0.56 ± 0.11 ^a^	0.74 ± 0.19 ^a^	0.77 ± 0.18 ^a^
C20:2	0.59 ± 0.17 ^a^	0.58 ± 0.19 ^a^	0.70 ± 0.14 ^a^	0.73 ± 0.10 ^a^
C20:3n-6	0.48 ± 0.12 ^a^	0.59 ± 0.14 ^a^	0.12 ± 0.02 ^a^	0.12 ± 0.02 ^a^
C20:4n-6	3.15 ± 0.78 ^a^	3.67 ± 0.57 ^a^	0.29 ± 0.07 ^a^	0.28 ± 0.07 ^a^
SFA	35.68 ± 2.09 ^a^	34.33 ± 1.64 ^a^	36.31 ± 1.93 ^a^	34.89 ± 2.88 ^a^
MUFA	44.58 ± 4.33 ^a^	45.91 ± 3.27 ^a^	46.47 ± 1.57 ^a^	47.86 ± 1.49 ^a^
PUFA	19.74 ± 3.11 ^a^	19.77 ± 2.66 ^a^	17.22 ± 1.66 ^a^	17.25 ± 2.09 ^a^
n-6	17.41 ± 3.88 ^a^	18.55 ± 2.86 ^a^	15.55 ± 2.48 ^a^	15.38 ± 2.22 ^a^
n-3	0.74 ± 0.25 ^a^	0.67 ± 0.19 ^a^	0.99 ± 0.19 ^a^	1.15 ± 0.13 ^a^
h	59.89 ± 2.08 ^a^	60.67 ± 1.51 ^a^	59.46 ± 2.05 ^a^	60.69 ± 2.05 ^a^
H	22.98 ± 1.78 ^a^	22.78 ± 0.75 ^a^	23.65 ± 1.19 ^a^	23.05 ± 1.67 ^a^

SFA: Saturated fatty acids; MUFA: monounsaturated fatty acids; PUFA: Polyunsaturated fatty acids; and h = hypocholesterolemic (sum of C18:1n-9, C18:1n-7, C18:2n-6, C18:3n-6, C18:3n-3, C18:4n-3, C20:3n-6, and C20:4n-6). H = Hypercholesterolemic (sum of C14:0 and C16:0); ^a^: Value in a row with different letters are significantly different (*p* < 0.05) (t de Student); equal letters for groups mean no differences for that characteristic.

## Data Availability

The data presented in this study are available on request from the corresponding authors.

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
