# Peer review of "Ligilactobacillus salivarius MP100 as an Alternative to Metaphylactic Antimicrobials in Swine: The Impact on Production Parameters and Meat Composition"

_animals, 2023, doi:10.3390/ani13101653_

Round 1

Reviewer 1 Report

Interesting study, but I do have questions on the statistical evaluation and presentation of data. Material: sows are F1-crossbred, but what about the boars (AI ?). Same breed of the boars during both periods?

Difficult to follow the process of the statistical analysis. 

Only t-test?  Also, it does not seem as you have taken into account the structure of the data. For the fertility traits, you will sure have repeated observations (one sow does most often show up 2-5 times during a 2-year period). I would strongly recommend you to redo the stat. analyses, according to a statistical model including (for the fertility traits): fixed effects of time period, parity number, year-month combination and the random effect of sow. Suggested stat. model for the fattening traits, since I suppose that you don't have litter identity on the pigs: Include fixed effects of: time period, gender, year-month combination. Proc mixed in the SAS package would handle this with no problems. 

Fig 1 to 3: to me it is most logic to have the 'pre-bar' to the left..... 

Author Response

Interesting study, but I do have questions on the statistical evaluation and presentation of data. Material: sows are F1-crossbred, but what about the boars (AI ?). Same breed of the boars during both periods?

Answer:
We agree that this is a relevant information that was not included in the original manuscript. In the case of boars, the same breed/genetics were used in both periods. The sows were artificially inseminated with semen from a TN Tempo Large White boar throughout both study periods. In the revised manuscript, we have included the following sentence (p. 2, l. 94-95):
“The sows are artificially inseminated with semen from TN Tempo Large White boars (AIM, Vught, The Netherlands) since 2013.”

Difficult to follow the process of the statistical analysis. Only t-test? Also, it does not seem as you have taken into account the structure of the data. For the fertility traits, you will sure have repeated observations (one sow does most often show up 2-5 times during a 2-year period). I would strongly recommend you to redo the stat. analyses, according to a statistical model including (for the fertility traits): fixed effects of time period, parity number, year-month combination and the random effect of sow. Suggested stat. model for the fattening traits, since I suppose that you don't have litter identity on the pigs: Include fixed effects of: time period, gender, year-month combination. Proc mixed in the SAS package would
handle this with no problems. 

Answer:
It is true that our data has a rather complex structure, such as repeated swine’ births over time and the presence of random effects. In these cases, the use of more advanced statistical models, such as mixed models or mixed effects models, may be appropriated but, in general, the use of t-tests to compare data in the same period (2 full years) and by batches (with the same management and technicians over the time) may be suitable as well. Despite some monthly variability was observed, we have tried to keep the variation of the variables at a minimum when comparing the two 2-years periods, in order to have a global view of the effect of changing a single management procedure (antimicrobials replaced by a probiotic strain). In order to cope with the reviewer’s comment, we have added the following paragraph in the
revised manuscript (p. 10, l. 388-397): “The statistical analysis of the high amount of complex data obtained in this study was a challenging issue because of some of the intrinsic characteristics of the assay, including the
recurrence of swine births over time and the potential occurrence of random effects. Although advanced statistical models, such as mixed models or mixed effects models, are suitable in such scenarios, our approach relied on the use of t-tests to compare data obtained during both 2-year periods under similar meteorological and management conditions with the only exception of the replacement of antimicrobials by a probiotic strain. Although we tried to keep variability at a minimum, the occurrence of random factors affecting the interpretation of our data cannot be excluded and this is one of the main limitations of this study.

”Fig 1 to 3: to me it is most logic to have the 'pre-bar' to the left....

Answer:
The revised manuscript has been corrected according to the reviewer’s comment.

Reviewer 2 Report

L58-81 - this is not an introduction or a presentation of the literature, but a description of the study. It shouldn't be here. This requires a complete rework. In this section you should present the literature on the impact of such products on production results, meat quality, or mention that such things have not been done.

L82-86 - the formulated goals seem incorrect. The title of the work is not the same as the goals of the work. And certainly the part I quoted: “first, to isolate and characterize a probiotic candidate from milk of an in-house sow with a good record of reproductive outcomes; second, to apply the strain as strategy to replace routine metaphylactic use of antibiotics in the farm where the strain was isolated; third, to evaluate the microbiological and biochemical impact of that replacement strategy.” The subject of the work is the impact on production results and meat quality. This needs to be changed.

L91: what boars covered the sows? This is very important in terms of obtaining meat. Because different boars can affect the quality of the pork harvested.

L111-115: did the housing conditions, nutrition, genetic material and other factors, such as weather conditions, change in the period from 2015 to 2019?

L121: were meat quality tests carried out on 6 pieces? It is not possible to count the stat correctly with this amount. Too small research group. Three animals from two groups cannot give information about the whole group

L122: what was the sex of the slaughtered animals?
Figure 1 a, b - it may be worth considering a better graphical representation of these graphs. In the case of 1b, the division of the scale every 5 is not clear to the recipient.

Figure 2 - similar note as in figure 1

L226-228: and how is daily weight gain measured? There is no information on how this parameter was controlled on the farm.

L234: what was the slaughter weight of the fatteners that were selected for the experiment?

L251: please replace SAFA with SFA and similarly in the table

L260: please replace "Latissimus" with "Longissimus"

L324: where are the meat pH results?

L329-331: "but intramuscular fat content and subcutaneous fat thickness were greater in the probiotic group, a fact that is in agreement with a higher growth rate." I don't fully agree with this statement. Certainly, the growth rate does not affect the thickness of subcutaneous fat.

L333-334: what was the composition of the compound feed in these two groups? Has oil been added to it? Were the genetics different?

Author Response

L58-81 - this is not an introduction or a presentation of the literature, but a description of the study. It shouldn't be here. This requires a complete rework. In this section you should present the literature on the impact of such products on production results, meat quality, or mention that such things have not been done.

Answer
We agree that the Introduction should be corrected to present the literature on the impact of such products on production and meat quality results. As a consequence, this section has been modified and a new paragraph has been included in the revised manuscript (p. 2, l. 56-65):
“In relation to probiotics, it has been reported that the administration of some lactobacilli strains may provide beneficial outcomes in performance [8-11] and meat quality [11-13]; however, only a few strains has been tested in real farm conditions. In addition, studies with other probiotic strains, mainly belonging to the genus Bacillus, have failed to find a positive impact or even have reported a detrimental effect on such parameters [14]. Such conflicting results may reflect differences in the properties of the administered strains and, therefore, a proper characterization of efficacy- and safety-related traits has a paramount relevance when selecting probiotic strains targeting swine production.
In this context, we have described recently the probiotic potential of a Ligilactobacillus salivarius strain isolated from sow’s milk [15].”
Accordingly, the references numbering and list have been modified in the revised manuscript.

L82-86 - the formulated goals seem incorrect. The title of the work is not the same as the goals of the work. And certainly the part I quoted: “first, to isolate and characterize a probiotic candidate from milk of an in-house sow with a good record of reproductive outcomes; second, to apply the strain as strategy to replace routine metaphylactic use of antibiotics in the farm where the strain was isolated; third, to evaluate the microbiological and biochemical impact of that replacement strategy.” The subject of the work is the impact on production results and meat quality. This needs to be changed.

Answer
We agree that the goals formulated in the paragraph cited by the reviewer is incorrect. As a consequence, such paragraph has been deleted in the revised manuscript.

L91: what boars covered the sows? This is very important in terms of obtaining meat. Because different boars can affect the quality of the pork harvested.

Answer

We agree that this is a relevant information that was not included in the original manuscript. In the case of boars, the same breed/genetics were used in both periods. The sows were artificially inseminated with semen from a TN Tempo Large White boar throughout both study periods.

In the revised manuscript, we have included the following sentence: “The sows are artificially inseminated with semen from TN Tempo Large White boars since 2013.” (p. 2, l. 94-95).

L111-115: did the housing conditions, nutrition, genetic material and other factors, such as weather conditions, change in the period from 2015 to 2019?

Answer

Housing conditions, genetic material and nutrition were essentially the same during the two 2-years periods. In fact, no major changes in such factors have been introduced in the farm since 2013. In relation to the weather conditions, although some differences were observed among the different months and years, the mean temperatures and pluviometry values in each season were very similar when both 2-years period were compared. Mean daily temperature values in the antibiotic period and probiotic period were, respectively: 11.2 and 11.5ºC in spring; 20.2 and 20.5ºC in summer; 14,3 and 14,1ºC in autumn and 5,5 and 5,3ºC in winter. In relation to pluviometry, the mean daily pluviometry values were as follows: 150.4 and 151.3 mm in spring; 91.1 and 87.4 mm in summer; 164,9 and 160.2 mm in autumn and 115.3 and 118.7 mm in winter.

This information has been included in the revised manuscript (p. 8-9, l. 310-321):

“It must be highlighted that no major changes have been introduced in this farm, in relation to housing conditions, genetics and nutrition since 2013; therefore, the impact of these factors when comparing the two 2-years periods was probably low. In relation to the weather conditions, although some differences were observed among the different months and years, the mean temperatures and pluviometry trimestral values were very similar when both 2-years period were compared (SiCLIMA; Sistema Básico de Información Climática de Aragón). More specifically, the mean daily temperature values in the antibiotic period and probiotic period in the farm location were, respectively: 11.2 and 11.5ºC in spring; 20.2 and 20.5ºC in summer; 14,3 and 14,1ºC in autumn and 5,5 and 5,3ºC in winter. In relation to pluviometry, the mean daily pluviometry values were, respectively, as follows: 150.4 and 151.3 mm in spring; 91.1 and 87.4 mm in summer; 164,9 and 160.2 mm in autumn and 115.3 and 118.7 mm in winter.”

L121: were meat quality tests carried out on 6 pieces? It is not possible to count the stat correctly with this amount. Too small research group. Three animals from two groups cannot give information about the whole group

Answer

A total of 6 samples (three from the probiotic group and three from the control group) have been analyzed in this study. Initially, it may seem a too small number of samples but it is not the case having in mind the methodology used and the objective pursued in this study. The objective was to carry out a metabolite profile analysis of the meat from both batches. This study is part of previous experiences of some of the co-authors of this work, in which the metabolite profiles of different meats and meat exudates have been studied (Castejon et al., 2015), including those of porcine origin (Garcia-Garcia et al. al, 2019). In all the cases in which the exudate of fresh pork meat was studied, the same metabolites have been found (see previous works by García-García et al., 2019 and Table S1). NMR spectroscopy is a robust, reliable and validated technology, and to know the metabolite profile it is not necessary to carry out repeated analysis of samples. This work reveals that probiotic intake does not alter the metabolomic profile of meat. When comparing the spectra of the samples from the control group and the probiotic group, the only difference to highlight was the presence of inosine that was only detected in the exudate samples corresponding to the pigs of the probiotic group.

In other words, the objective of this work was not to carry out a quantitative study, an approach that would require the corresponding statistical analysis to check the variability between samples, but a qualitative approach to elucidate if the use of probiotics may alter the profile of metabolites. In previous works (Castejón et al., 2015; Garcia-Garcia et al., 2019), it has been observed that IMP and inosine signals disappear with the spoilage and aging of meat. Therefore, IMP and inosine signals seem to be associated to meat freshness.

Regarding the fatty acid profile, again the individual analysis of each sample revealed that no significant differences were observed in the quantitative analysis of each fatty acid as well as in the calculated indices.

So, although studies involving a higher number of animals and samples are required to confirm our findings (and, in fact, it has been recognized in the manuscript), we think that our analyses serve, at least, to show that the administration of the probiotic strain did not exert a negative impact on meat quality. Therefore, we would like to keep this part in the revised manuscript (if possible).

L122: what was the sex of the slaughtered animals?

Answer

We have included this information in the revised manuscript (castrated male pigs).

Figure 1 a, b - it may be worth considering a better graphical representation of these graphs. In the case of 1b, the division of the scale every 5 is not clear to the recipient.

Answer

The Figure has been amended according to the reviewer’s comment.

Figure 2 - similar note as in figure 1

Answer

The Figure has been amended according to the reviewer’s comment.

L226-228: and how is daily weight gain measured? There is no information on how this parameter was controlled on the farm.

Answer

We have included this information in the revised manuscript (p. 3, l. 115-117):

“To measure the daily weight gain, the weight of each animal at slaughter was divided by the number of days from its birth.”

L234: what was the slaughter weight of the fatteners that were selected for the experiment?

Answer

We have included this information in the revised manuscript (~110 kg).

L251: please replace SAFA with SFA and similarly in the table

Answer

The manuscript has been amended according to the reviewer’s comment.

L260: please replace "Latissimus" with "Longissimus"

Answer

The manuscript has been amended according to the reviewer’s comment.

L324: where are the meat pH results?

Answer

The pH values have been included in Table 1.

L329-331: "but intramuscular fat content and subcutaneous fat thickness were greater in the probiotic group, a fact that is in agreement with a higher growth rate." I don't fully agree with this statement. Certainly, the growth rate does not affect the thickness of subcutaneous fat.

Answer

We agree with the reviewer’s comment and, as a consequence, this sentence has been eliminated in the revised manuscript.

L333-334: what was the composition of the compound feed in these two groups? Has oil been added to it? Were the genetics different?

Answer

We have included this information in the revised manuscript (p 3, l. 128-133):

“For this purpose, three commercial castrated male pigs from the study farm were compared to three equivalent individuals from a control farm that used the same feed but, also, a standard metaphylactic antimicrobial treatment. The feed was elaborated in Cooperativa Alto Aragón de Barbastro SCL (Barbastro, Spain) and contained wheat, barley, corn, bran soybeans, beet pulp, lard (3.9%) and vitamin and mineral correctors.”

Round 2

Reviewer 1 Report

OK, accept

Author Response

Dear reviewer,

Thank you very much for your time and effort in reviewing this article.

Reviewer 2 Report

Many thanks to the authors for the corrections made.

As it stands, I believe the paper meets the requirements for publication in the journal Animals.

Please change SAFA to SFA in L266 and Table 2.

In addition, I still do not fully agree that three samples from each group is a result that speaks of the entire population of tested pigs. If other reviewers and the editor of the journal will admit him to further processes, I agree. For me, this is too small a number of trials to talk about entire populations.

Author Response

Dear Reviewer,

Thank you for thinking that the revised paper meets the requirements for publication in the journal Animals. We understand that you still do not fully agree that the sample size is enough to speak of the entire population of tested pigs. However, we sincerely think that it is important to include these results in the manuscript as a kind of proof-of-concept since our objective was to check that the intake of the strain did not negatively affect the composition/quality of the meat. We agree that additional studies are required to confirm the findings (and this limitation has been acknowledged in the manuscript) but these preliminary results will give us the opportunity to get funding for an in depth characterization of a much larger sample size and to include other complementary techniques to assess meat quality and composition. The other reviewer has endorsed the revised manuscript for publication and we hope it will be the editor’s decision. Anyway, we value your comments and criticisms since they will help us to improve our future works.

Please change SAFA to SFA in L266 and Table 2.

Done
